# Quantification of Residual Stress Relief by Heat Treatments in Austenitic Cladded Layers

**DOI:** 10.3390/ma15041364

**Published:** 2022-02-12

**Authors:** Joana Rebelo Kornmeier, Maria José Marques, Weimin Gan, António Castanhola Batista, Sanjooram Paddea, Altino Loureiro

**Affiliations:** 1MLZ-Heinz Maier-Leibnitz Zentrum, Technische Universität München, 85748 Garching, Germany; 2University of Coimbra, CFisUC, Department of Physics, 3004-516 Coimbra, Portugal; mjvaz@fe.up.pt (M.J.M.); castanhola@uc.pt (A.C.B.); 3Department of Physics Engineering, University of Porto, R. Dr. Roberto Frias, 4200-465 Porto, Portugal; 4German Engineering Materials Science Centre (GEMS) at MLZ, Helmholtz-Zentrum Geesthacht GmbH, 85748 Garching, Germany; weimin.gan@hzg.de; 5Faculty of MCT, The Open University, Walton Hall, Milton Keynes MK7 6AA, UK; s.paddea@open.ac.uk; 6Centre of Excellence for Advanced Materials, Dongguan 523808, China; 7University of Coimbra, CEMMPRE, Department of Mechanical Engineering, Rua Luis Reis Santos, 3030-788 Coimbra, Portugal; altino.loureiro@dem.uc.pt

**Keywords:** residual stresses, neutron diffraction, contour method, stainless steel cladding, heat treatment

## Abstract

The effect of the heat treatment on the residual stresses of welded cladded steel samples is analyzed in this study. The residual stresses across the plate’s square sections were determined using complementary methods; applying diffraction with neutron radiation and mechanically using the contour method. The analysis of the large coarse grain austenitic cladded layers, at the feasibility limits of diffraction methods, was only made possible by applying both methods. The samples are composed of steel plates, coated on one of the faces with stainless steel filler metals, this coating process, usually known as cladding, was carried out by submerged arc welding. After cladding, the samples were submitted to two different heat treatments with dissimilar parameters: one at a temperature of 620 °C maintained for 1 h and, the second at 540 °C, for ten hours. There was some difference in residual stresses measured by the two techniques along the surface of the coating in the as-welded state, although they are similar at the welding interface and in the heat-affected zone. The results also show that there is a residual stress relaxation for both heat-treated samples. The heat treatment carried out at a higher temperature showed sometimes more than 50% reduction in the initial residual stress values and has the advantage of being less time consuming, giving it an industrial advantage and making it more viable economically.

## 1. Introduction

Process vessels are widely used in many diverse industries such as power, petroleum, or chemical industries. For economic reasons, the cladding of the internal surfaces of ferritic steel process vessels is a common practice to avoid corrosion and provide the necessary mechanical properties as detailed in [1]. Frequently, the ferritic steel vessels are cladded with stainless steel weld beads. Automatic submerged arc welding (SAW) is one of the most widely used methods of making cladded surfaces because of its high quality and reliability [2]. However, as a consequence of the welding process, in addition to the formation of hard and brittle martensitic structures [3], the surrounding base material is subject to complex thermomechanical cycles that include elastic, plastic, and/or creep distortions. Welding processes normally create huge residual stresses of a tensile nature close to the weld zone. The residual tensile stresses may promote during service operation, due to fatigue, the growth of cracks generated in brittle structures during welding [4]. As stated by Suárez et al. [4], the amount and distribution of the residual stresses of cladding are influenced by: the history of the temperature throughout the cladding process parameters, the geometry and thickness of the cladded component, the mismatch of the physical and mechanical characteristics of the cladding and base metal, and the post-weld heat or mechanical treatments. Tensile residual stresses in engineering components, such as process vessels, degrade structural performance, particularly when manufacturing or service-induced faults are present. These vessels commonly exhibit cracking at the metal/cladding interface, affecting their uses. According to [5], fracture initiation is greatly influenced by the local properties at the cladding’s interfaces.

Carbon migration to austenitic stainless steel during welding and post-weld heat treatment is one of the main causes of cracking, which is increased if the material contains hydrogen and residual tensile stresses [6]. The service life of components operating at high temperatures and pressures is significantly harmed by residual tensile stresses. As a consequence, it is critical to evaluate and anticipate the amount and spatial extent of these residual stresses as part of an engineering structural integrity assessment program.

Diffraction techniques are well-suited and commonly used in the determination of macroscopic residual stresses associated with welding. Diffraction techniques are phase sensitive, and using neutrons instead of lab X-rays (XRD) has the advantage of allowing strains to be measured for most polycrystalline metals or ceramics non-destructively at depths of several millimeters through the samples rather than being confined to the surface. Although XRD is widely used due to its high spatial resolution, reliability and easier accessibility, but it is often semi-destructive in thickness measurements or suffers the detrimental effect of the grain structure [7]. Neutron diffraction is a purely non-destructive method (assuming stress-free samples can be extracted from a second sample or quantified in a different way) and usually possess enough spatial resolution (of the order of millimeters) compared with the residual stress length-scale in welds. Unfortunately, neutron diffraction is only available at large-scale facilities, which makes it less accessible than lab XRD. Nevertheless, both diffraction methods may still be very localized, especially when coarse grain materials must be analyzed, such as in this study, concerning welds of austenitic steels, with dendrites spanning several millimeters. Large grain size or the crystallographic texture of the material make it difficult to quantify residual stresses, either by XRD [8] or using neutron diffraction [9]. Another big issue using the neutron diffraction method is the determination of the reference value. Daymond and Johnson [10] calculated the reference value using elastic anisotropy, Ganguly et al. [11] showed the limitations of using the well-known comb measurements and Withers et al. [12] discussed different methods and good practices to choose the reference value in different situations. Moreover, Brown et al. [13] showed the need to use both methods due to the type II or intergranular stresses in orthorhombic materials such as uranium. With respect to cladded components, the neutron diffraction method is likely to average out any variation owing to weld bead variations [14]. The well-known pattern of the “innate scatter” of residual stress regions in welds makes the depiction of residual stresses a difficult task.

An attempt to study residual stress profiles in welds using an artificial neural network approach was made in [15]. Although this scattering of results is often attributed to the texture of the dendrites, it has been shown that texture does not play a key role in austenitic steel stress analysis [9]. The contour method established by Prime [16] is an excellent alternative/complementary method to diffraction methods for residual stresses assessments. This method consists of measuring surface deformations resulting from stress relaxation induced by a cutting plane and is based on the Bueckner’s elastic superposition principle. This method allows drawing 2D maps of residual stresses perpendicular to the cutting plane. The contour and neutron diffraction methods rely on completely different assumptions. Since the contour method premises on macroscopic mechanical relaxation, it is, therefore, indifferent to material features on the atomic level. Zhang et al. [17] showed that the contour method is very suitable for evaluating residual stresses in welds at high thicknesses.

The effectiveness of the post-weld heat treatments used in industry to relieve the stress in austenitic cladded layers and ferritic base material is analyzed and discussed in this study. Two different industrial heat treatments were studied and compared with the as-welded state (AW). The state of biaxial residual stresses from the surface across the interface of the cladded austenite up to the ferritic base material were analyzed by diffraction and mechanical means, using neutron radiation and the contour method respectively. Two different spatial profiles, positioned at the TOP and BOTTOM of a weld pass, were acquired in the middle of each plate using neutron diffraction in the usual three main directions. Those are longitudinal, transverse, and normal to the weld pass. The TOP and BOTTOM in depth profiles were calculated using anisotropic elasticity constants derived from neutron texture measurements [9]. The residual stress profiles were compared and discussed in relation to the ones calculated by Marques et al. [14] using isotropic elasticity constants. Contour maps for the two main weld directions, longitudinal and transverse of the individual samples were acquired. The respective TOP and BOTTOM residual stress profiles for the longitudinal direction obtained by neutron diffraction and the contour method were compared and discussed.

## 2. Materials

In the present study, three austenitic steel layers were applied over a ferritic base plate of 20 mm thick, with the submerged the submerged arc welding (SAW) process, according to Figure 1, using a wire diameter of 4 mm and a heat input of about 1.2 kJ/mm. Details of the base and cladding materials are listed in Table 1 and the SAW parameters in Table 2. The first cladding layer was constituted of AISI 309L whilst AISI 316L was used for the second and third layers, each cladding layer had approximately 2.5 mm thick. This is general practice to avoid abrupt compositional changes due to the dilution of the parent materials and electrode, in order to prevent cracking. The coating procedure was carried out on a single sheet (500 × 500 mm^2^), and later cut into four parts by EDM, in order to obtain samples of 250 × 250 mm^2^. This procedure was adopted in order to guarantee that the coating conditions were identical in all samples. Two of the samples were subjected to two commonly used industrial heat treatments, one at 620 °C and the other at 540 °C, with holding times of 1 h and 10 h, respectively. The selection of heat treatment conditions was based on industrial requirements. From now on, the three types of samples in this study will be denoted as follows: as-welded (AW) and HT620 and HT540, for the heat-treated samples described above, respectively.

## 3. Methods for Measuring Residual Stress

### 3.1. Neutron Diffraction

The diffraction of a neutron beam irradiating a crystalline sample, by a group of crystal lattice planes {hkl}, with lattice spacing *d_hkl_*, occurs when the magnitude of the length $2d_{hkl}\sin\theta_{hkl}$ equals an integer multiple of the respective used wavelength. The scattering angle of *2θ_hkl_* is determined by Bragg’s law.
(1)$$n\cdot \lambda =2\cdot d_{hkl}\cdot \sin\theta_{hkl}$$
where *n* is an integer indicating the order of the diffraction phenomenon. Only the elastic strain change in the lattice spacing Δ*d_hkl_* = (*d_hkl_* − *d_0,hkl_*) contributes to the diffraction of the neutron beam, resulting in an angular shift Δ2*θ*_hkl_. The differentiation of Bragg’s equation guides the elastic lattice strain component, *ε_hkl_* parallel to the scattering vector, which is normal for the lattice planes {hkl}. The *d_0,hkl_* is the reference lattice parameter, which corresponds to the unstressed material state.
(2)$$\varepsilon_{hkl}=\left(d_{hkl}-d_{0,hkl}\right)/d_{0,hkl}$$

When performing strain measurements by neutron diffraction, strain scanners are used. Neutron strain scanners are specialist instruments found at large-scale facilities, where non-destructively spatially resolved measurements, concerning different sample scans, can be carried out by defining a specific volume of material which is irradiated, the so-called gauge volume. Gauge volumes can be specified precisely by primary and secondary optic systems, depending on what is under investigation. In this study, strain and localized texture measurements were carried out using the neutron material diffractometer STRESS-SPEC at FRM II, Technische Universität München (TUM), in Garching, Munich. STRESS-SPEC is a strain scanner dedicated to the analysis of the residual stress of materials and is characterized in [18]. Texture measurements and analysis are described in [19]. For the present strain measurements, the horizontally bent silicon Si (400) monochromator was used to provide a monochromatized neutron beam with a wavelength of approximately 1.667 Å. A monochromator takeoff angle of around 2θ_M_ = 76° was chosen resulting in scattering angles of roughly 2θ_S_ ≈ 91.4° and of 2θ_S_ ≈ 101° for the ferrite and austenite sample phases respectively, see Figure 2. For the longitudinal direction, where a higher spatial resolution is required, a gauge volume of 2 × 2 × 2 mm^3^, was defined by conventional slits, whereas for the transverse and normal directions a bigger and time saving gauge volume of 2 × 10 × 2 mm^3^ was set up, as described in the primary study conducted by Marques et al. [9].

Considering that the principal directions could be deduced from the weld geometry as assumed in [20], the strain measurements in the three directions were sufficient to evaluate the residual stresses. Two through-thickness strain profiles were acquired in the usual main weld directions, the so-called longitudinal, transverse, and normal directions to the weld pass as displayed in Figure 1. The above-mentioned strain profiles were acquired in the central point (TOP) and at the edge (BOTTOM) of one weld pass, as indicated by the red dots in Figure 1. Reference measurements were measured in stress-free samples, 3 × 3 × 3 mm^3^ cubes, Figure 1, machined by wire electro discharge machining (EDM), from duplicate samples at the same depths of the measurements of the respective cladded plates. Detailed description of the measurements can be seen at [14].

For the local texture measurements, 10 mm diameter cylindrical rods were machined from a duplicate cladded plate by EDM, Figure 1. For the texture measurements another STRESS-SPEC setup was used. The Ge (311) monochromator was set at a take-off angle of 58.45° to produce a monochromatic neutron beam with a wavelength of approximately 1.666 Å.

A 3 mm diameter circular slit in the primary beam and a radial collimator with a full width at half maximum (FWHM) of 2 mm on the diffracted beam side defined the gauge volume, as shown in Figure 3.

Detailed information about the texture measurements, procedure and data analysis were reported in the study by Rebelo-Kornmeier et al. [9]. Part of the neutron diffraction results were already published [9,14] for instance, but never recalculated or compared with those of other techniques.

### 3.2. Contour Method

The contour method is grouped in the mechanical methods for measuring residual stresses. It is destructive in nature since it involves making a cut in the component of interest, on the plane in which the residual stresses are to be evaluated. Its main principle is based on the relaxation of the residual stresses that existed within the component previous to the cut, causing the cut surfaces to deform. As detailed in [11], the morphology of the cut surfaces was precisely assessed experimentally, and finite element (FE) modelling was performed to establish the state of the crosswise residual stresses existing previous to the cut.

In the present work, the contour cuts, see Figure 4, were made by wire electro-discharge machining (WEDM). An Agie Charmiles machine was used and the cuts were made by a 0.15 mm diameter brass wire of sacrificial material, as detailed by Fischer et al. [21]. After the cutting by wire, the resulting surface deformation was quantified by a micro-Epsilon laser sensor on a 0.025 mm × 0.025 mm grid by Zeiss Eclipse coordinate measuring machines (CMM).

The data acquired were then treated according to the procedures that are detailed in the following description, and in the same order of priority. First, the alignment of the data measured from the cut surfaces, then the suppression of the noise and outliers, succeed by averaging the deformations of the matching cut surfaces in order to subtract the shear stress effects and cutting flaws. The final step of the data processing consists in data smoothing (using spline).

Afterwards, a three dimensional FE model of the cut part was created and appropriately meshed where the opposite of the measured smoothed deformations was applied as a boundary condition to the cut face. As a consequence, the corresponding residual stress in the normal direction was calculated from a linear elastic finite element analysis using ABAQUS software [22].

Residual stresses acting transversely to the direction of the cladding on the mid-width longitudinal plane of the sample, that are present prior to performing the 1st cut (measuring longitudinal stress), can be determined by using the principle of superposition, as described in [23], i.e., undisturbed transverse stresses = ‘relaxed’ transverse stresses + ‘remaining’ transverse stresses. The contour FE analysis of the first cut provides the ‘relaxed’ transverse stress distributions over the entire cut part. The ‘remaining’ transverse stresses can be determined from a second contour cut.

## 4. Results and Discussion

### 4.1. Neutron Diffraction

Neutron strain profiles were acquired for all the samples and principal strain directions, the longitudinal, transverse, and normal to the weld pass. The measurement setup and respective residual stress calculations that were carried out taking into consideration Young’s isotropic moduli are detailed in [9]. Isotropic elasticity constants, experimentally determined by Eigenmann and Macherauch [24], were also used in the present study. For the ferritic (Fe-α) lattice plane, {211}, the values of E_211_ = 220 GPa, ν_211_ = 0.28 and austenitic (Fe-γ) lattice plane {311}, the values of E_311_ = 175 GPa, ν_311_ = 0.31 were employed for the residual stress calculations respectively. The localized texture analysis conducted by Rebelo-Kornmeier et al. [9] was the basis for the calculations of the tri-axial global Young’s moduli of the cladded austenite for each sample. The analysis shows that fiber textures are developed after the welding process. Furthermore, the intensity level of the texture increases with depth. In addition, the texture effect is slightly higher after both heat treatments compared with the AW sample. The respective anisotropic elastic constants were than used to re-evaluate the residual stress profiles analyzed, taking into consideration the influence of texture. The triaxial global Young’s moduli were determined using the Cub_PHY program which was established on a cluster model as in the studies carried out by Kiewel et al. [25] and Park et al. [26]. By using the complete pole figures measured and the harmonic series expansion method, the C-coefficients were calculated through the orientation distribution functions (ODFs). The global Young’s moduli values determined of each sample are shown in Figure 5, continuously from the longitudinal direction (0°) until the transverse direction (90°). The normal directions calculated are summarized in Table 3.

The anisotropy of Young’s moduli is visible in all samples. The differences range from 5% to over 20%. After welding, for the AW sample, the anisotropy is the highest and is substantially reduced following the heat treatment HT540. For each sample the longitudinal direction always exhibits the lowest values. Although the heat treatments, in comparison with the AW sample, lead to an almost imperturbable texture effect, they also have a high impact by dampening the Young’s moduli anisotropy, particularly for HT540. This must be due to the dissolution of the delta (δ) phase during the heat treatment, as stated in the work performed by Vandermeulen et al. [27] and Gill et al. [28] and observed in the present samples [29]. The Young’s moduli values observed in this study agree very well with those earlier measured for identical steels carried out with XRD [13,27].

The impact of the Young’s moduli anisotropy measured on the respective residual stress values in depth were compared and plotted in the longitudinal direction at the TOP position, see Figure 6. Moreover, the BOTTOM residual stress profiles were plotted in the same graph for the different samples to show the much higher sensitivity of the residual stress profiles according to the measuring position of the sample, compared with the influence of anisotropy. The differences in residual stress values, considering the isotropic or anisotropic Young’s moduli, can vary between 5% and 10%, depending on the different positions and samples. These variations may be attributed to the presence of texture. In all of the samples studied, strong fiber textures are registered and which increase with depth [10]. This type of texture agrees very well with the ones measured at the surface with XRD [13] in similar cladded materials. Those residual stress differences are, however, much smaller than the ones observed for the TOP or BOTTOM profiles. These results show a strong dependence on the measuring positions, as they display a manifold and localized behavior. This behavior is very well-demonstrated in Figure 6 for the AW and HT540 samples. The slight differences in the residual stress values, when anisotropic Young’s modulus is used, compared with the ones calculated using isotropic elasticity constants, may be related to the much weaker fiber texture of the {311} diffraction planes on which the strain measurements are based. In Figure 7, pole figures of the evolution {311} of the diffraction lines with depth, for the distinct cladded layers for the three samples are plotted. These were calculated from the respective orientation distribution function (ODF) obtained after the complete pole figure measurements of the {111} and {200} diffraction lines of the different cladded layers, reported by Rebelo-Kornmeier et al. [9].

The {311} textures are significantly weaker than the textures observed for the {111} and {200} diffraction lines for all the samples and respective layers. This is why Hutchings et al. [30] recommend that strain measurements using diffraction techniques should be carried out on this lattice plane. It can be seen in Figure 7 that the minor texture variations, in depth, show exactly the same tendency which is observed concerning the slight differences in the residual stress values calculated using isotropic or anisotropic elastic constants. For instance, and for the AW sample, the texture is stronger at a greater depth as well as the difference in the residual stress values, Figure 6, mainly following the same tendency of the {311} texture. On the contrary, for the HT540 sample, the residual stress divergence, Figure 6, and texture are higher close to the surface, in the outer austenitic layer of the cladding. For the HT620 sample, a major divergence of the residual stress values is observed, see Figure 6, just as for the strength of the texture (Figure 7) in the second cladded layer.

Due to the diversity of the results, mainly related to the localized characteristic of the neutron diffraction method for those large grain materials, and since a neutron full-field two-dimensional map is time consuming, the contour method was also applied to provide greater confidence in the resulting assessment of the stress state.

### 4.2. Residual Stress Maps after Contour Method

The maps of the residual stress state measured for the understudy samples in the longitudinal and transverse directions are plotted in Figure 8, Figure 9 and Figure 10.

The contour method provides an overview of the stress distribution in the weld-coated sections highlighting, in particular, the influence of the weld seams in the creation of high longitudinal compressive stresses, in dark blue in Figure 8a, at the interface with the base material in the AW condition. Note that between these dark blue areas small red areas appear, indicating high tension stresses. The transverse residual stresses in sample AW, near the interface, are of lower value, but of tension. The contour maps shown above and the residual stresses obtained using neutron diffraction have the heterogeneity of the residual stress state in common, i.e., the form and amplitude of the profile in depth is very sensitive to the position analyzed for each plate. The contour maps show clearly, however, that overall, the residual stresses peak-to-valley amplitudes decrease in the longitudinal direction after the heat treatments particularly at the interface of the parent material with the first cladding layer. On the contrary, the amplitudes slightly increase in the perpendicular direction, both tensile and compression, see Figure 9b and Figure 10b.

The variations in the stress profiles with the positions on the plate can be seen clearly in Figure 11, Figure 12 and Figure 13 for each sample, AW, HT540 and HT620 respectively. The residual stress profiles in depth for similar topological positions, TOP, and BOTTOM, on the left, middle and right side of the plates are plotted together for the longitudinal direction and each sample.

These figures show substantial differences in the residual stresses measured in the middle of the weld seams (TOP) and in the boundary zones between weld seams (BOTTOM). In the top zones, residual tensile stresses of up to 300 MPa can be observed, close to the welding interface, while in the bottom zones, compressive stresses of up to 500 MPa can be observed in the AW condition. Heat treatments reduce the residual compressive stresses at the weld interface but increase the residual tensile stresses in the molten austenitic zone; however, the evolution of stresses in thickness is similar for the three samples.

### 4.3. Discussion

Assessment of the heat treatment after cladding is only possible following a discussion of the above results. Examples of the results of both methods are shown and compared in Figure 14 and Figure 15. The results are from the AW and HT620 samples for the same topology, TOP, and BOTTOM, in the middle plate positions.

Figure 14 shows that there are differences between the longitudinal residual stresses measured by the two methods, mainly in the areas close to the surface of the coated plate. These differences are more significant in the zone of interaction between adjacent welds (BOTTOM), Figure 14b, although they approximate in the zone of interface between the weld and the base material. The comparison between Figure 14a,b shows that the distribution of the welds influences the value and profile of the installed residual stresses. On the other hand, the unexpected very low tensile stresses for the AW sample, measured by the contour method, see Figure 14a,b, in the austenitic layers of the cladding can only be attributed to plasticity effects which may be present at the tip of the cut. The procedure associated with the contour method is highly sensitive to the quality of the sectioning cut, and mistakes linked to the plasticity produced by stress redistribution ahead of the WEDM wire during the cutting process have been witnessed in some situations [31,32]. One can infer that the material is plastically deformed in the cladded layers from the stress magnitudes calculated by neutron diffraction, see Figure 6. They are above the nominal yield strength of austenite, approximately 300 MPa. This effect does not appear under compression since the yield strength of austenitic steel under compression is much higher. The plastic effect is usually connected to the strain resulting from the phase transformations, such as austenite to martensite in stainless steel [33].

Disparities in the residual stress profiles are not only observed in different topological regions, as obtained by neutron diffraction, but also in different plate positions of similar topological weld regions, see Figure 11, Figure 12 and Figure 13. For instance, at the same depth and similar topological profile position, TOP or BOTTOM, there are variations from the maximum compression to maximum tension of the residual stress values. Generally, those differences are registered at the beginning of the interface region at around 7.5 mm, see detailed graphs in Figure 11, Figure 12 and Figure 13. The maximum values under compression reach amounts of almost −300 MPa and the maximum for the tension is between 150 and 200 MPa, for the heat-treated samples and almost −500 MPa and over 300 MPa respectively for the AW sample. To explain the observed residual stress increase, further research must be carried out in order to evaluate the eventually new mechanical properties of the material at the interface. The mechanical material properties may no longer correspond to the base or weld material. The compressive values close to the surface agree very well with the X-ray diffraction and the results of the hole drilling measurements (HDMs) for these samples [29].

For both samples and in the base material, the residual stress values show an excellent agreement between both methods. This is often the case as shown by various comparative studies of both methods [23,34,35,36,37]. Some differences occur at the interface between the parent ferritic steel and austenitic cladded layers. For the AW sample, see Figure 14, the residual stress values acquired from the neutron diffraction are in general higher than those obtained by the contour method. This has been recognized by different authors and is related to the microstructure of the materials, such as steel materials [38] and magnesium alloys [39].

For the heat-treated sample, the stress profiles acquired by both methods, contour and neutron diffraction, see Figure 15, agree reasonably well in stress amplitudes and profiles. The main discrepancy is an approximately constant displacement of the residual stress values; an offset is observed. This may be due to the fact that the plasticity effects were not considered in the calculations for the contour method as stated in [40,41]. Although the effects of material plasticity are challenging to predict, since they are strongly correlated to the prior processing the material has undergone, the associated mistakes were reduced by safely clamping the plates during the cutting [16].

It has been shown that the results of mechanical based methods match the diffraction methods very well and provide complementary data, which would otherwise only be available using less accessible neutron and synchrotron radiation. This is clearly shown in studies of residual stress states by mechanical post weld treatments of steels [42] or in the studies of where the characterization of the residual stresses of low transformation temperatures of alloyed steels occurred [43]. Other studies have shown the limits of using neutron diffraction in very thick steel specimens [42,44], which were only made possible by using different wavelengths to go through to the residual stress state. Complex sample shapes, such as the T shape can also be a disadvantage when using diffraction methods [45]

The contour method and its dissimilar assumptions to the diffraction methods makes it an excellent alternative and complementary method to assess residual stresses, particularly in welds. The residual stresses present in a sample analyzed, using the contour method, are evaluated by the total elastic deformations after the stress is relieved by the cut. Thus, the huge and heterogenic austenitic grains present in the cladded layers, characterized for the present samples by [29], do not affect the assessment of the residual stress as they affect diffraction methods. Furthermore, the contour method makes it possible to map the residual stresses present over all the cross-sections at once with one cut. Thus, as the contour method provides a global picture, it is more suited to assess the post heat-treatment effects on the stress state after cladding processes for coarse grain materials. The certainty of the quantified residual stress relieved by the heat treatment is made only evident by using both methods. Although there are some justified discrepancies among the results obtained from the procedures of this complementary analysis, this study shows that the contour method corroborates the neutron diffraction results. The main disadvantage of the contour method is that it is not always possible to apply it, as it is destructive.

## 5. Conclusions

The contour method, in addition to neutron diffraction, is used to assess the effect of heat treatment on the residual stresses of large grain austenitic cladded layers on ferritic base plates in the present work. The former, although destructive in nature, provides a more accurate overview of the tendencies shown by the neutron diffraction results regarding the complete cross section. The latter, despite the fact that it is non-destructive, has only a limited use for coarse grain materials as it provides very localized results. There was some difference in residual stresses measured by the two techniques along the surface of the coating in the as-welded state, although they are similar at the welding interface and in the heat-affected zone. The cutting method used in the contour method can contribute to this difference.

Results also show that there is a residual stress relief after both heat treatments of the cladded plates, not just at the point where the cladded layers meet the base material, but also in the outer layers close to the surface of the cladding. There does not seem to be a clear advantage to any of the heat treatment procedures in terms of relieving residual stresses, although the HT620 treatment is more economically favorable, due to the shorter time required.

## Figures and Tables

**Figure 1 materials-15-01364-f001:**
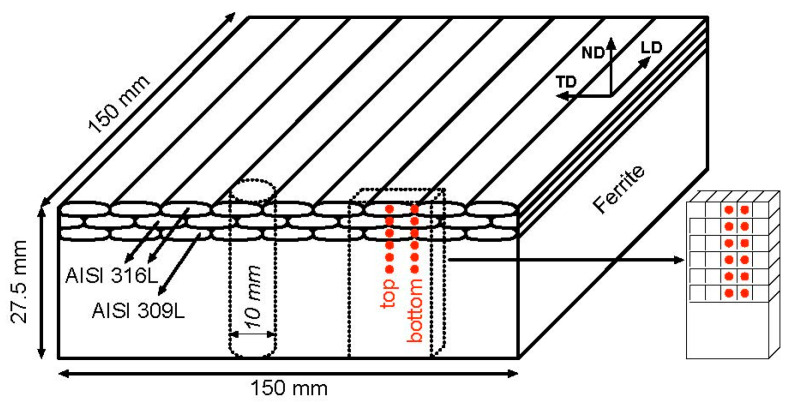
Duplicate cladded plate where: dimensions, materials, measurement positions and directions; stress free and local texture sample schemas are shown.

**Figure 2 materials-15-01364-f002:**
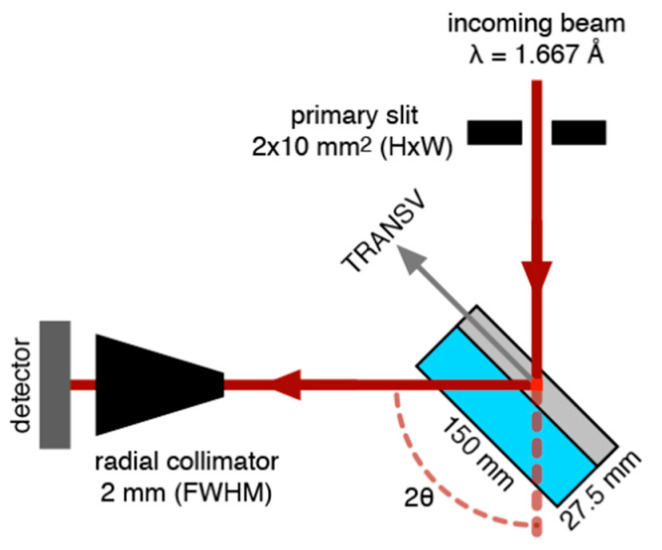
Neutron diffraction setup schema where an example of measurements for transverse direction is shown.

**Figure 3 materials-15-01364-f003:**
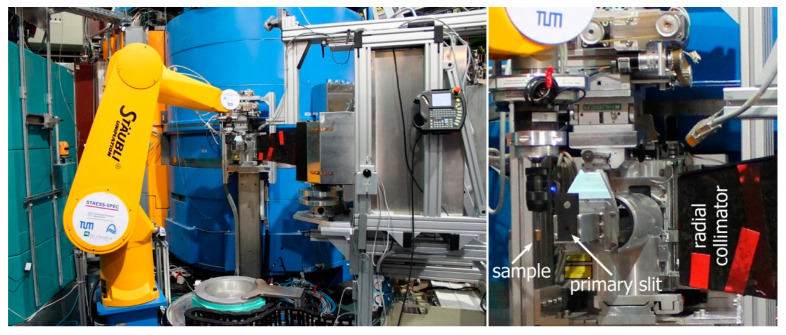
On the left, the STRESS-SPEC robot used in lieu of the conventional translation table plus a Eulerian cradle as the instrument setup for local texture measurements. On the right, details of the sample’s primary and secondary optic system are shown.

**Figure 4 materials-15-01364-f004:**
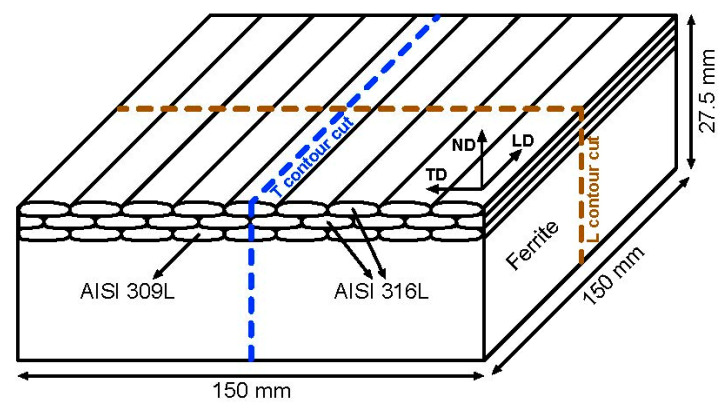
Schematic representation of the contour cuts of the cladded samples.

**Figure 5 materials-15-01364-f005:**
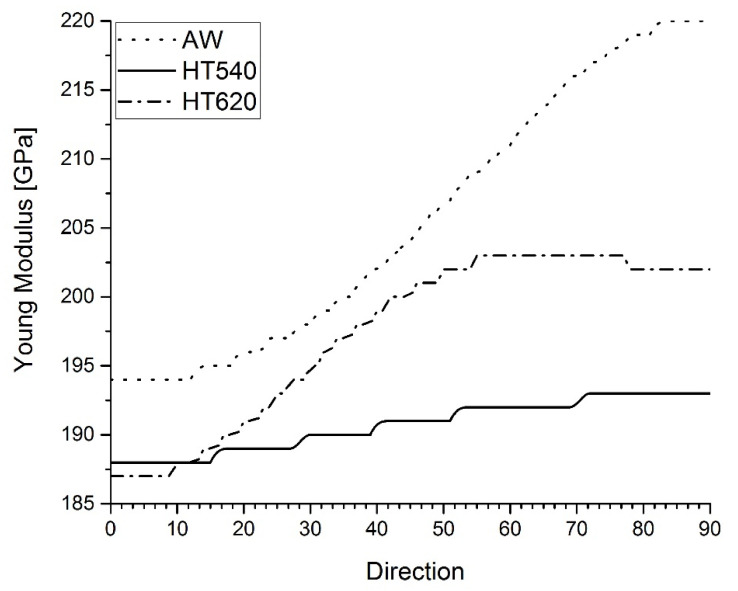
Calculated global Young’s moduli of AW, HT540, and HT620 from longitudinal (0°) to transverse direction (90°).

**Figure 6 materials-15-01364-f006:**
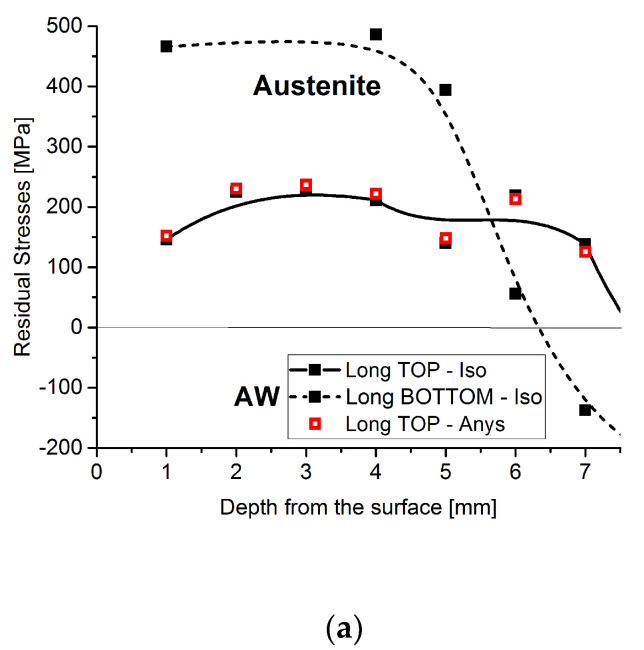
Evolution of longitudinal residual stress in depth of the austenitic cladded layers at the TOP and BOTTOM positions for each sample: (**a**) AW, (**b**) HT540, and (**c**) HT620. For comparison, the residual stress values for the TOP positions using anisotropic elastic constants are also shown, in legend shown by Long TOP-Anys.

**Figure 7 materials-15-01364-f007:**
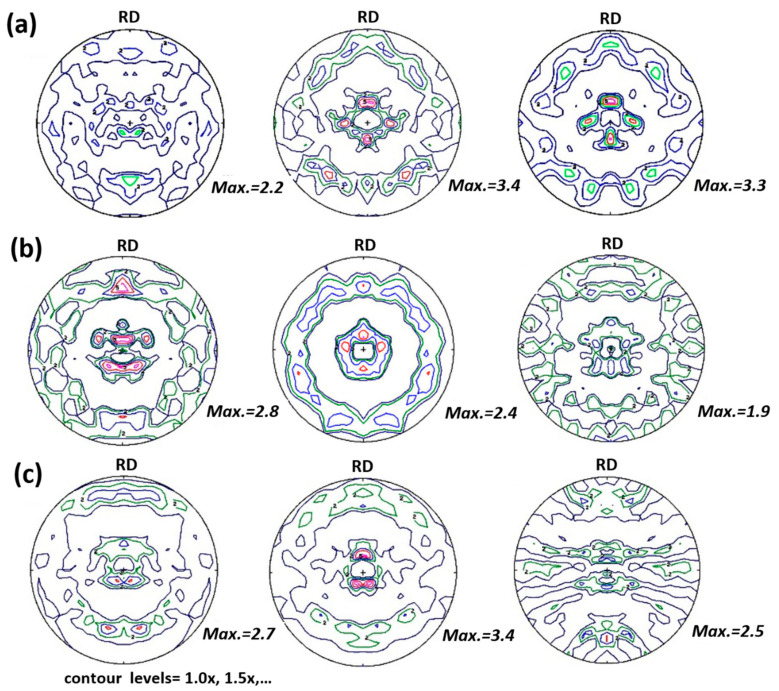
Calculated pole figures in depths of the diffraction plane {311} for (**a**) as weld sample (AW) and heat-treated samples (**b**) HT540 and (**c**) HT620. For (**a**–**c**) the left-hand pictures were measured at 2 mm depth, the middle ones at 5 mm depth and the pictures on the right-hand side were acquired at 7.5 mm depth.

**Figure 8 materials-15-01364-f008:**
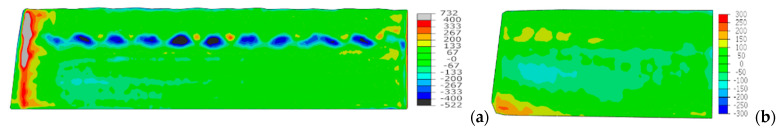
AW residual stress maps for the (**a**) longitudinal and (**b**) transverse directions.

**Figure 9 materials-15-01364-f009:**
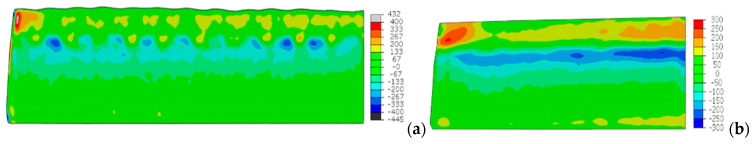
HT540 residual stress maps for the (**a**) longitudinal and (**b**) transverse directions.

**Figure 10 materials-15-01364-f010:**
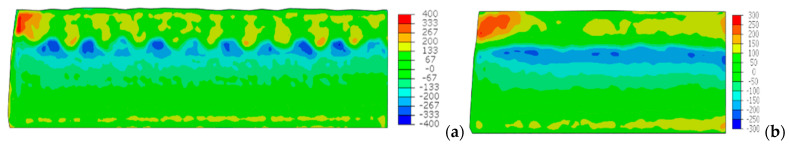
HT620 residual stress maps for the (**a**) longitudinal and (**b**) transverse directions.

**Figure 11 materials-15-01364-f011:**
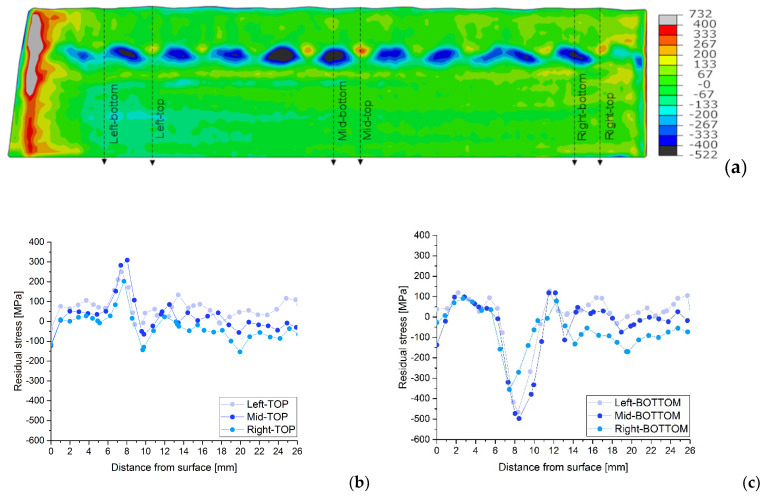
Longitudinal residual stress map for AW (**a**) and profile details at selected TOP (**b**) and BOTTOM (**c**) positions, indicated in the respective maps.

**Figure 12 materials-15-01364-f012:**
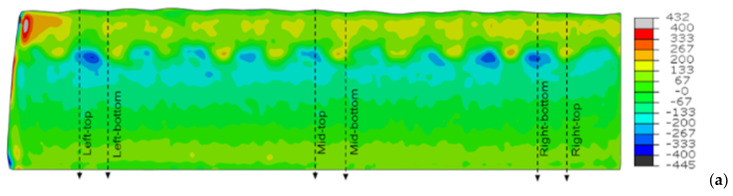
Longitudinal residual stress map for HT540 (**a**) and profile details at selected TOP (**b**) and BOTTOM (**c**) positions, indicated in the respective maps.

**Figure 13 materials-15-01364-f013:**
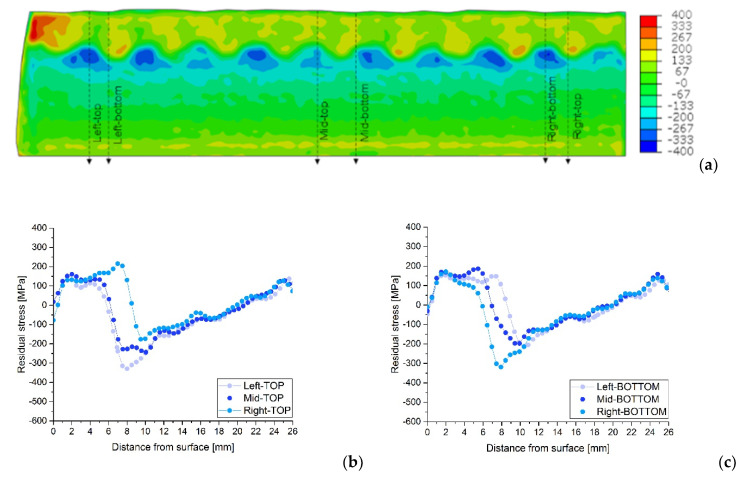
Longitudinal residual stress map for HT620 (**a**) and profile details at selected TOP (**b**) and BOTTOM (**c**) positions, indicated in the respective maps.

**Figure 14 materials-15-01364-f014:**
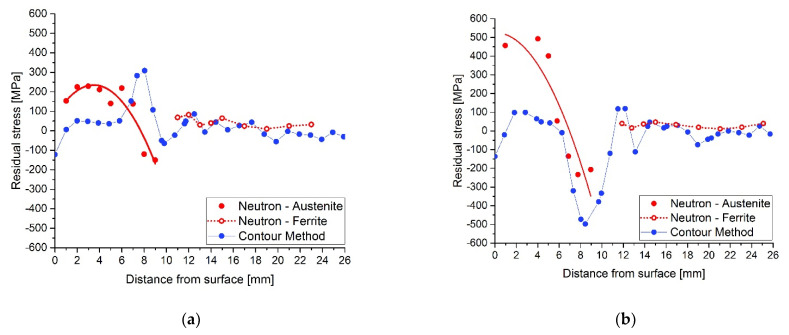
Comparisons of the longitudinal residual stress profiles for the AW sample obtained by both methods at (**a**) TOP and (**b**) BOTTOM weld positions in the middle of the plate.

**Figure 15 materials-15-01364-f015:**
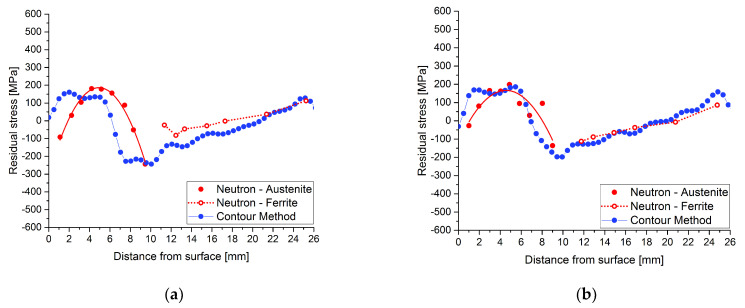
Comparisons of the longitudinal residual stress profiles for the HT620 sample obtained by both methods at (**a**) TOP and (**b**) BOTTOM weld positions in the middle of the plate.

**Table 1 materials-15-01364-t001:** Chemical composition of the substrate and cladded layers, in wt.% (balance Fe) [14].

Material Name	C	Si	Mn	S	Cr	Ni	N	P	Cu	Mo
Parent material (P355 NH)	0.18	0.33	1.12	0.002	0.04	0.16	0.004	0.015	0.21	-
1st cladding layer (AISI 309L)	0.008	0.29	1.4	0.003	21.3	14.8	0.057	0.016	0.11	2.57
2nd and 3rd cladding layers (AISI 316L)	0.018	0.38	1.8	0.009	18	11.7	0.055	0.016	0.07	2.54

**Table 2 materials-15-01364-t002:** Flux composition used in the SAW and principal process parameters [14].

Flux Composition (wt.%)				
SiO_2_	MgO	CaF_2_	Al_2_O_3_	Cr
31	25	6	13	4.8
Arc voltage		29–30 V DC (+)		
Arc current		300 A		
Preheating		150 °C		

**Table 3 materials-15-01364-t003:** Young’s moduli (GPa) determined for the normal directions of each sample and the three cladded layers.

	AW	HT540	HT620
Third cladded layer	201	153	158
Second cladded layer	171	146	162
First cladded layer	205	178	181
global	183	151	165

## Data Availability

The raw/processed data required to reproduce these findings cannot be shared at this time as the data also forms part of an ongoing study.

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
