# Peer review of "Quantification of Residual Stress Relief by Heat Treatments in Austenitic Cladded Layers"

_materials, 2022, doi:10.3390/ma15041364_

Round 1

Reviewer 1 Report

Journal: Materials

Manuscript ID: materials-1578769

Title: Quantification of residual stress relief by heat treatments in
austenitic cladded layers

In the present work, effect of the heat treatment on the residual stresses of welded cladded steel samples is analyzed. Two different methods were applied to measure the stresses. The heat treatment carried out at a higher temperature showed sometimes more than 50% reduction of the initial residual stress values. The work shows reasonable novelty and useful to automotive and mechanical applications. The paper can be accepted in Journal “Materials’ after clarifying the below given points.

  1. Author need to provide the reason for selecting this experimental design. Author must offer scientific reasons for your choice that have been used?
  2. There is some coherence missing in the introduction part; need to revise the introduction part to show the clear research gap/flow. Provide some more related literature.
  3. The nomenclature must contain a complete list of symbols, markings, abbreviations - it must be at the beginning of paper. Please include it in the next version of the paper.
  4. 6, Fig.11-15 quality need to be improved to publication standard.
  5. Reference 15, it’s not 115. Please correct it.
  6. If possible, author can provide the picture of various cladded /heat treated samples for more clarity.
  7. The hardness property will play a greater role with/without residual stress. Why author did not perform the indentation test.

Author Response

1 - The coating procedure was carried out on a single sheet (500x500 mm), and later cut into four parts by EDM, in order to obtain samples of 250x250 mm. This procedure was adopted in order to guarantee that the coating conditions were identical in all samples. For clarity we decided to include this sentence in the manuscript.

2 - The Reviewer is right, the elaboration of an introduction to this article is very complex, as it requires the presentation of very different information to the reader. We confess that we tried several organizations, but in the end the one that seemed most coherent was the one we presented in the manuscript. We begin by presenting the reasons for making these coatings and their industrial interest. Below we highlight the consequences of this process, both in metallurgical terms and in terms of residual stresses, showing the importance of measuring stresses. Neutron diffraction and contour measurement methods are also presented and their main advantages and limitations are described. Finally, it is mentioned that the objective is to compare the results obtained with the two techniques, taking into consideration new principles adopted in the case of neutron diffraction.

3 - The list of symbols and abbreviations is usual in articles with a mathematical character, which is not the case. All symbols and abbreviations are explained in the text, placed in their context. We are afraid that some, when taken out of context, lose meaning. However, if the reviewer considers it essential, the list can be made.

4 - Unfortunately, it is very difficult to improve the quality of images in time.

5 - The reference was corrected.

6 - Images of the weld layers are illustrated in references [9, 14, 29], so their presentation in the current article was considered unnecessary, as the samples are the same.

7 - The hardness profiles are detailed in the references [9, 29], so the authors think that their presentation in the current manuscript is unnecessary.

The authors are grateful for the reviewer's help in improving the document.

Reviewer 2 Report

Please find detailed evaluation in the attached file.

Author Response

Fig. 11-13: Please double check the positions ………

Response - The residual stress profiles positions were chosen at the plates to emphasize the diversity of the results one can get. Two at the left, two at the middle and two at the right side of the plate. In every spatial region, left, middle or right we choose one at the bottom and the other at the top of the weld bed positions. We were focused on showing the profile differences of the results in each plate.

General

Response - Regarding the designations top and bottom, they were used intentionally, to show that the morphology of the coating, ie the coating procedure, may influence the stress fields. The designation Top refers to a line drawn in the middle of a weld bead of the upper layer, the highest area of the coating, while Bottom refers to a line drawn in the area of interaction between two surface welds in a row, the morphologically lower area, as is illustrated in Figure 2. The simple use of the labels "Line 1" and "Line 2" can hide this intention.

Page 3

Response - Dear reviewer, the objective of the article is to compare the residual stress fields measured by neutron diffraction, using new data treatment principles, with those measured by the contour method, obtained on as-welded and heat-treated samples. Although this is stated in the text, I agree with the reviewer that the introduction on line 115 of the previous sentence helps to clarify the objectives.

Page 14

Response - A further sentence has been added to the discussion:

To explain the observed residual stress increase, further research has to be carried out in order to evaluate the eventually new mechanical properties of the material at the interface. The mechanical material properties may no longer corresponds to the base or weld material.

Minor comments

Page 2

Response - The font size of the word Although has been changed and the reference number corrected.

Page 3

Response - Figure 2 appears now first, and is renumbered as figure 1.

The reason for using the AISI309 electrode in the first layer is related to the fact that it is a heterogeneous bond between carbon steel and stainless steel. This requires that the electrode be chosen so that the chemical composition of the molten metal, given the dilution of the base metals and electrode, falls into a specific area of the Schaeffler diagram, in order to avoid different types of cracking. This is an eminently metallurgical topic, which departs from the purpose of the article. In any case, the authors decided to introduce the following sentence into the manuscript:

This is a general practice to avoid abrupt compositional changes due to the dilution of the parent materials and electrode, in order to prevent cracking.

Page 4

Response - Figure 1 is renumbered as figure 2.

The authors are grateful for the reviewer's help in improving the document.